

# Volatile anesthetics suppress glucose-stimulated insulin secretion in MIN6 cells by inhibiting glucose-induced activation of hypoxia-inducible factor 1

Kengo Suzuki[1], Yoshifumi Sato[2], Shinichi Kai[1], Kenichiro Nishi[1], Takehiko Adachi[3], Yoshiyuki Matsuo[1] and Kiichi Hirota[1]

[1] Department of Anesthesiology, Kansai Medical University, Hirakata, Osaka, Japan
[2] Department of Medical Biochemistry, Faculty of Life Sciences, Kumamoto University, Kumamoto, Japan
[3] Department of Anesthesia, Tazuke Kofukai Medical Research Institute Kitano Hospital, Osaka, Japan

## ABSTRACT

Proper glycemic control is one of the most important goals in perioperative patient management. Insulin secretion from pancreatic $\beta$-cells in response to an increased blood glucose concentration plays the most critical role in glycemic control. Several animal and human studies have indicated that volatile anesthetics impair glucose-stimulated insulin secretion (GSIS). A convincing GSIS model has been established, in which the activity of ATP-dependent potassium channels ($K_{ATP}$) under the control of intracellular ATP plays a critical role. We previously reported that pimonidazole adduct formation and stabilization of hypoxia-inducible factor-1$\alpha$ (HIF-1$\alpha$) were detected in response to glucose stimulation and that MIN6 cells overexpressing HIF-1$\alpha$ were resistant to glucose-induced hypoxia. Genetic ablation of HIF-1$\alpha$ or HIF-1$\beta$ significantly inhibited GSIS in mice. Moreover, we previously reported that volatile anesthetics suppressed hypoxia-induced HIF activation *in vitro* and *in vivo*. To examine the direct effect of volatile anesthetics on GSIS, we used the MIN6 cell line, derived from mouse pancreatic $\beta$-cells. We performed a series of experiments to examine the effects of volatile anesthetics (sevoflurane and isoflurane) on GSIS and demonstrated that these compounds inhibited the glucose-induced ATP increase, which is dependent on intracellular hypoxia-induced HIF-1 activity, and suppressed GSIS at a clinically relevant dose in these cells.

## INTRODUCTION

Proper glycemic control is one of the most important goals in perioperative patient management (*Lipshutz & Gropper, 2009*; *Martinez, Williams & Pronovost, 2007*). A series of studies has clearly demonstrated that hyperglycemia represents a serious risk factor for perioperative morbidity and mortality (*Kwon et al., 2013*; *Lipshutz & Gropper, 2009*). Insulin secretion from pancreatic $\beta$-cells in response to an increase in the blood glucose concentration plays a critical role in glycemic control.

Corresponding author
Kiichi Hirota, hif1@mac.com

Several animal and human studies have indicated that volatile anesthetics such as halothane, enflurane, isoflurane, and sevoflurane impair insulin secretion in response to glucose administration. The blood glucose concentration reflects an elaborate balance between the generation and utilization of glucose; this is determined by the secretion of, and the resistance to, insulin. This internal balance, however, can be disturbed by external factors such as surgical insults and the drugs used for anesthetic management.

A convincing model of glucose-induced insulin secretion has been established, based on considerable experimental evidence (*Rorsman, 1997*; *Seino, 2012*). There is a consensus that the intracellular ATP concentration ([ATPi]) plays a crucial role in GSIS. When the extracellular glucose concentration increases, pancreatic $\beta$-cell metabolism accelerates, leading to an increase in [ATPi]. As a result of these metabolic changes, the activity of ATP-dependent potassium channels ($K_{ATP}$) decreases; this causes membrane depolarization to the threshold potential at which voltage-dependent calcium channels open, allowing $Ca^{2+}$ influx. The ensuing increase in cytosolic $Ca^{2+}$ concentration triggers exocytosis of insulin-containing vesicles. We and others have demonstrated that glucose induced a high level of $O_2$ consumption (*Kurokawa et al., 2015*), resulting in intracellular hypoxia that was strong enough to activate the hypoxia-inducible factors (HIF), HIF-1 and HIF-2, in pancreatic $\beta$-cells (*Bensellam et al., 2012*; *Sato et al., 2011*). A series of *in vivo* studies has demonstrated that volatile anesthetics, including halothane, isoflurane, sevoflurane, and desflurane suppressed GSIS, resulting in a dysregulation of the blood glucose concentration (*Lipshutz & Gropper, 2009*; *Martinez, Williams & Pronovost, 2007*; *Tanaka et al., 2011a*). In addition, several reports indicated that these anesthetics disturbed GSIS by inhibiting $K_{ATP}$ closure (*Tanaka et al., 2009*). Genetic ablation of the HIF-1$\alpha$ or HIF-1$\beta$ gene was shown to reduce GSIS in mice (*Cheng et al., 2010*; *Gunton et al., 2005*; *Pillai et al., 2011*). Moreover, we previously reported that halothane, sevoflurane, and isoflurane suppressed hypoxia-induced activation of HIFs *in vitro* and *in vivo* (*Itoh et al., 2001*; *Kai et al., 2014*; *Tanaka et al., 2011b*).

The present series of experiments examined the effects of volatile anesthetics (sevoflurane and isoflurane) on GSIS and demonstrated that both anesthetics inhibited the glucose-induced increase in [ATPi], which is dependent on intracellular hypoxia-induced HIF-1 activity, and suppressed GSIS at a clinically relevant dose in the mouse MIN6 insulinoma pancreatic $\beta$-cell line.

## MATERIALS AND METHODS

### Reagents

Isoflurane was obtained from Dainippon Sumitomo Pharma Co., Ltd. (Osaka, Japan) and sevoflurane was from Maruishi Pharmaceutical Co., Ltd. (Osaka, Japan). Oxygen (Taiyo Nippon Sanao, Wakayama, Japan), and nitrogen (Taiyo Nippon Sanso, Tokyo, Japan) were also used. An inhibitor of HIF, 5-[1-(phenylmethyl)-1H-indazol-3-yl]-2-furanmethanol (YC-1), the HIF$\alpha$ hydroxylase inhibitor, dimethyloxaloylglycine (DMOG), the selective $K_{ATP}$ (Kir6 subunit) blocker, glibenclamide, and the activator, diazoxide, were all obtained from Abcam (Cambridge, MA, USA). n-Propyl gallate (nPG; 3,4,5-trihydroxybenzoic acid

propyl ester), the mitochondrial uncoupler, carbonyl cyanide m-chlorophenylhydrazone (CCCP), sucrose, and maltose were all obtained from Sigma Aldrich (St. Louis, MO, USA).

## Cells and cell culture

The mouse insulinoma MIN6 and MIN7 cell lines were a gift from Dr. J Miyazaki (Osaka University) (*Miyazaki et al., 1990*). MIN6 and MIN7 cells were maintained at 37 °C under 5% $CO_2$ and 95% air in Dulbecco's modified Eagle's medium (DMEM) (Gibco, Grand Island, NY, USA) containing 450 mg/dl glucose, 10% fetal bovine serum (FBS), penicillin, streptomycin, and 50 μM $\beta$-mercaptoethanol.

## Treatment with volatile anesthetics

Cells were maintained in an airtight chamber or work-station (AS-600P; AsOne, Osaka, Japan) perfused with mixed air (MODEL RK120XM series; Kofloc, Kyotanabe, Japan) with or without each test anesthetic, delivered by a specialized vaporizer within the open circuit. The concentrations of gases and anesthetics in the incubator were monitored during each treatment using an anesthetic gas monitor (Type 1304; Blüel & Kjær, Nærum, Denmark) that was calibrated with a commercial standard gas (47% $O_2$, 5.6% $CO_2$, 47% $N_2O$, 2.05% sulfur hexafluoride) (*Itoh et al., 2001*). The anesthetic concentration in the medium was measured by gas chromatography (5890A; Hewlett Packard, Palo Alto, CA, USA), as described previously (*Namba et al., 2000*).

## Measurement of insulin concentration

Insulin secretion into the culture medium from MIN6 cells was measured using the Mouse Insulin H-type$^{TM}$ enzyme-linked immunosorbent assay kit (Shibayagi Co. Ltd., Shibukawa, Japan), according to the manufacturer's protocol (*Matsumoto et al., 2012*). Briefly, MIN6 cells were cultured for 30 min in Krebs-Ringer bicarbonate HEPES (KRBH) buffer (140 mM NaCl, 3.6 mM KCl, 0.5 mM $NaH_2PO_4$, 0.5 mM $MgSO_4$, 1.5 mM $CaCl_2$, 2 mM $NaHCO_3$, 10 mM HEPES, 0.1% bovine serum albumin) containing 40 mg/dl glucose. Then, the KRBH buffer was changed to one containing the indicated concentration of glucose, and the cells were cultured for 10 min or 1 h. The KRBH buffer was collected and subjected to insulin measurement. The cells on the dish were washed once with phosphate-buffered saline, lysed, and collected by scraping. The cells were sonicated and the protein concentration was determined using a protein assay kit (BioRad Laboratories, Hercules, CA, USA), with bovine serum albumin as the standard. The insulin concentration of the buffer was normalized to the total cell protein level. The results were normalized to the concentration of control samples of each independent experiment and the normalized values were demonstrated as insulin secretion ratio.

## Cytotoxicity assay

Changes in the cellular viability of MIN6 cells were determined by the CellTiter 96$^{TM}$ AQueous One Solution cell proliferation assay (Promega, Madison, WI, USA) (*Kai et al., 2012*). The assay uses a colorimetric method to determine the number of viable cells in cytotoxicity assays. MIN6 cells were seeded in 96-well plates at a density of $2.0 \times 10^4$ per

well (in 100 μl medium). After 24 h, cells were exposed to isoflurane or sevoflurane for 8 h. MTS ([3-(4,5-dimethylthiazol-2-yl)-5-(3-carboxymethoxyphenyl)-2-(4-sulfophenyl)-2H-tetrazolium, inner salt]/phenazine ethosulfate (PES) solution was added and incubation was continued for 30 min. The absorbance of individual wells was then measured at a wavelength of 490 nm corrected to 650 nm using a Thermo Max$^{TM}$ microplate reader (Molecular Devices, Sunnyvale, CA, USA). Assays were performed at triplicate at least twice. Data were expressed as mean ± standard deviation (SD).

## Assessment of apoptosis in MIN6 cells

Apoptosis was measured with the Apo-ONE$^{TM}$ Homogeneous Caspase-3/7 Assay (Promega) according to the manufacturer's protocol. The assay contains proflourescent rhodamin 110 (Z-DEVD-R110) which serves as a substrate for both Caspase-3 and -7. MIN6 cells were seeded in 96-well plates at a density of $2.0 \times 10^4$ per well (in 100 μl medium). After 24 h, cells were exposed to isoflurane or sevoflurane. After an additional 8 h of incubation, fluorescence was measured with Ensopire$^{TM}$ plate reader (PerkinElmer, Waltham, MA, USA) to determine the Caspase-3/7 activity. Assays were performed at triplicate at least twice. Data were expressed as mean ± standard deviation (SD).

## Measurement of total cellular O$_2$ consumption (OCR)

The total OCR was measured as described previously (*Kai et al., 2012*; *Zhang et al., 2007*). Cells were trypsinized and suspended at $1 \times 10^7$ cells per ml in DMEM containing 10% FBS and 25 mM HEPES buffer. For each experiment, equal numbers of cells suspended in 1 ml were pipetted into the chamber of an Oxytherm electrode unit (Hansatech Instruments, Norfolk, United Kingdom), which uses a Clark-type electrode to monitor the dissolved O$_2$ concentration in the sealed chamber over time. The data were exported to a computerized chart recorder (Oxygraph; Hansatech Instruments) that calculated the OCR. The temperature was maintained at 25 °C during measurement. The O$_2$ concentration in 1 ml of DMEM medium without cells was also measured over time to provide background values. O$_2$ consumption experiments were repeated at least thrice. Data were expressed as mean ± standard deviation (SD) (*Kai et al., 2012*). MIN6 cells were pre-exposed to the volatile anesthetics isoflurane and sevoflurane for 1 h. Cells were harvested and OCR measurement was performed in a working chamber. CCCP was added just before OCR measurement.

## Immunoblot assays

Whole-cell lysates were prepared as described previously (*Goto et al., 2015*; *Tanaka et al., 2010*). In brief, these were prepared using ice-cold lysis buffer (0.1% SDS, 1% Nonidet P-40 [NP-40], 5 mM EDTA, 150 mM NaCl, 50 mM Tris-Cl [pH 8.0], 2 mM DTT, 1 mM sodium orthovanadate, and Complete Protease Inhibitor$^{TM}$ (Roche Diagnostics, Tokyo, Japan)) using a protocol described previously (*Kai et al., 2012*). Samples were centrifuged at $10,000 \times g$ to sediment the cell debris, and the supernatant was used for subsequent immunoblotting experiments. For HIF-1$\alpha$ and HIF-1$\beta$ determinations, 100 μg of protein was fractionated by sodium dodecyl sulfate-polyacrylamide gel electrophoresis (7.5% gel),

transferred to membranes and immunoblotted using primary antibodies at a dilution of 1:1,000. Horseradish peroxidase-conjugated to sheep anti-mouse IgG (GE Healthcare, Piscataway, NJ, USA) was used as a secondary antibody at a dilution of 1:1,000. The signal was developed using enhanced chemiluminescence reagent (GE Healthcare, Little Chalfont, UK). Experiments were repeated at least two times and the representative blots were demonstrated.

## Measurement of [ATPi]

MIN6 cells were plated in a 96-well tissue culture plate. After the indicated treatments, [ATPi] was determined using a Cellno ATP Assay Kit (TOYO BNet, Tokyo, Japan), according to the manufacturer's instructions. Briefly, 100 μl of the lysis/assay solution provided by the manufacturer was added to the cells. After shaking for 1 min and incubating for 10 min at 23 °C, the luminescence of an aliquot of the solution was measured in a luminometer (ExSpire™, Perkin Emler, Waltham, MA, USA) (*Koyanagi et al., 2011*).

## Quantitative reverse transcriptase-PCR analysis

RNA was purified using RNeasy™ (Qiagen, Valencia, CA, USA) and treated with DNase. First-strand synthesis and real-time PCR were performed using the QuantiTect SYBR green PCR kit (Qiagen), according to the manufacturer's protocol. PCR primers were purchased from Qiagen. PCR and detection were performed using a 7300 real-time PCR system (Applied Biosystems, Foster City, CA, USA). The relative change in expression of each target mRNA relative to 18S rRNA was calculated (*Suzuki et al., 2013*).

## Gene silencing using short interfering RNA (siRNA)

siRNAs corresponding to mouse HIF-1$\alpha$ were from Qiagen Inc. MIN6 cells were transfected by 100 nM siRNA using HiPer-Fect™ Transfection Reagent (Qiagen) following a protocol provided by the manufacturer (*Oda et al., 2008*).

## Statistical analysis

All experiments were repeated on at least two occasions in triplicate. Data were expressed as the mean $\pm$ SD and analyzed by one-way analysis of variance, followed by Turkey's multiple comparisons test. All statistical analyses were performed with EZR (Saitama Medical Center, Jichi Medical University), which is a graphical user interface for R (The R Foundation for Statistical Computing, version 3.1.3) (*Kanda, 2013*). More precisely, it is a modified version of R commander (version 1.6–3) and includes statistical functions that are frequently used in biostatistics. A $P$-value of $<0.05$ was considered statistically significant.

# RESULTS

## Establishment of the GSIS experimental system in MIN6 cells

First, we investigated the insulin secretion response of MIN6 and MIN7 cells to stimulation by extracellular glucose. MIN6 and MIN7 cells were maintained with 40 mg/dl glucose and then exposed to a range of glucose concentrations (40–400 mg/dl) for 1 h. Concentration-dependent GSIS was observed in response to 100–400 mg/dl glucose in MIN6 cells
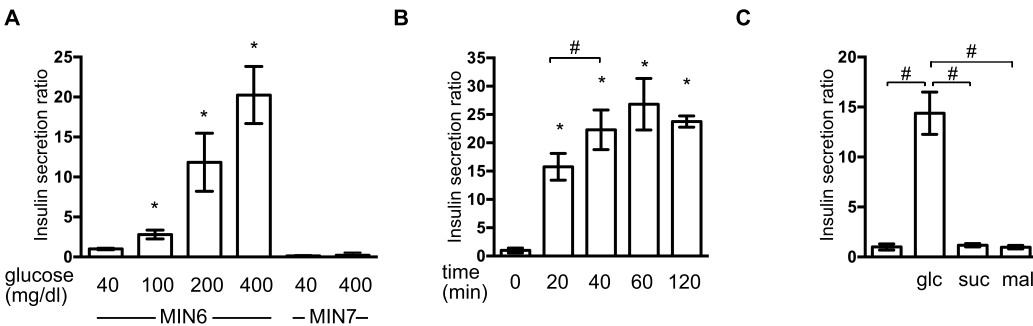

**Figure 1 Establishment of the GSIS experimental system in MIN6 cells.** (A) MIN6 and MIN7 cells were stimulated with the indicated glucose concentrations for 1 h and insulin secretion was determined as the difference between the medium insulin concentration before and after stimulation, as described in 'Materials and Methods.' (B) MIN6 cells were incubated with 400 mg/dl glucose for the indicated times prior to calculation of insulin secretion. (C) MIN6 cells were stimulated by 400 mg/dl (20 mM) glucose (glc), 20 mM sucrose (suc), or 20 mM maltose (mal) for 1 h prior to calculation of insulin secretion. Data are presented as the mean ± SD ($N = 3$, $n = 8$); *$P < 0.05$, as compared with the control (A: glucose 40 mg/dl, B: time 0 min); #$P < 0.05$ for comparison of the indicated groups.

(Fig. 1A). GSIS was not observed in MIN7 cells (Fig. 1A) (*Miyazaki et al., 1990*). The time profile of the GSIS of MIN6 cells was also examined. The cells were exposed to medium containing 400 mg/dl glucose and the insulin concentrations were measured at 10, 20, 40, 60, and 120 min (Fig. 1B). Insulin secretion reached its maximum point at 40 min. The insulin secretion response to the polysaccharides, sucrose and maltose, was investigated at molar concentrations corresponding to 400 mg/dl glucose (Fig. 1C). Insulin secretion was only observed in the presence of glucose in MIN6 cells.

## Reversible inhibition of GSIS by isoflurane and sevoflurane

We examined the effect of isoflurane and sevoflurane on GSIS in MIN6 cells. The cells were incubated with the indicated dose of anesthetic and 400 mg/dl glucose for 1 h prior to determination of the insulin concentrations of the culture supernatants. Isoflurane and sevoflurane inhibited GSIS significantly in a concentration-dependent manner between 0.6% and 2.4% for isoflurane (Fig. 2A), and between 0.8% and 3.6% for sevoflurane (Fig. 2B).

To examine whether this suppression of GSIS was reversible, MIN6 cells were exposed to 1.2% isoflurane or 1.8% sevoflurane with 400 mg/dl glucose for 1 h; they were then incubated under 40 mg/dl glucose without isoflurane or sevoflurane for 6 h, prior to re-exposure to 400 mg/dl glucose. No statistically significant differences were observed in the GSIS of MIN6 cells pretreated with volatile anesthetics and those that were not exposed to these compounds (Fig. 2C).

## The effect of volatile anesthetics on molecular aspects of GSIS in MIN6 cells

To investigate the molecular mechanisms underlying volatile anesthetic-mediated GSIS inhibition, cell proliferation and cell death were examined in MIN6 cells. Exposure to these volatile anesthetics (isoflurane: 2.4%, sevoflurane: 3.6%) for 8 h did not induce caspase

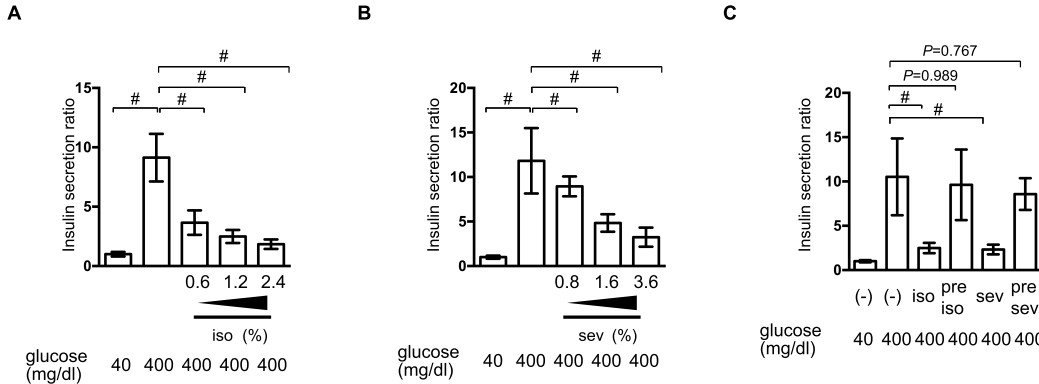

**Figure 2 Reversible inhibition of GSIS by isoflurane and sevoflurane.** (A) and (B) MIN6 cells were stimulated with the indicated glucose concentrations for 1 h, with or without the indicated dose of isoflurane (iso) (A) or sevoflurane (sev) (B), prior to calculation of insulin secretion. (C) MIN6 cells were exposed to 400 mg/dl glucose and the indicated concentrations of isoflurane or sevoflurane for 1 h, then incubated with 40 mg/dl glucose without isoflurane or sevoflurane for 6 h, and then exposed to 400 mg/dl glucose conditions for 1 h prior to calculation of insulin secretion. Data are presented as the mean ± SD (A and B: $N = 3$, $n = 10$, C: $N = 2$, $n = 6$); #$P < 0.05$ for comparison of the indicated groups. $N$, number of independent experiments performed; $n$, number of samples.

3/7 activation in MIN6 cells (Fig. 3A) and 8-h exposure periods did not affect the cell proliferation rate (Fig. 3B). This indicated that exposure to these volatile anesthetics did produce any statistically significant effects on MIN6 cell death or proliferation.

The expression of proteins involved in GSIS was investigated. mRNA expression of the glucose transporter 2 (GLUT2), the Kir6.2 subunit of $K_{ATP}$, and the voltage-dependent calcium channel, Cav1.2, was not affected by glucose or by the volatile anesthetics within 1 h (Fig. 3C).

## The effect of volatile anesthetics on intracellular signaling processes involved in GSIS

[ATPi] has been reported to increase in response to high-glucose stimulation in pancreatic $\beta$-cells (*Ashcroft, 2005*). We investigated the effect of isoflurane and sevoflurane over intracellular ATP concentration at 40 min and 2 h after exposure to 400 mg/dl glucose. In MIN6 cells, the mitochondrial electric transfer chain inhibitor, rotenone (100 nM), suppressed the glucose-induced increase in [ATPi]. Exposure to 2.4% isoflurane or 3.6% sevoflurane suppressed the increase in [ATPi] observed in response to 400 mg/dl glucose stimulation at 40 min (Fig. 4A) and 2 h (Fig. 4B). But the effects were weaker than 100 nM rotenone, which suppressed GSIS (Fig. 4C).

Elevation of [ATPi] closes $K_{ATP}$ and depolarizes the plasma membrane (*Ashcroft, 2005*; *Seino, 2012*). The channel opener, diazoxide, inhibited insulin secretion in pancreatic $\beta$-cells (*Seino, 2012*), while channel blockade by glibenclamide facilitated insulin secretion, even under low-glucose conditions. In MIN6 cells, glibenclamide induced insulin secretion in the presence of 400 mg/dl glucose, but even with 40 mg/dl glucose; conversely, diazoxide suppressed GSIS observed in the presence of 400 mg/dl glucose (Fig. 4D).

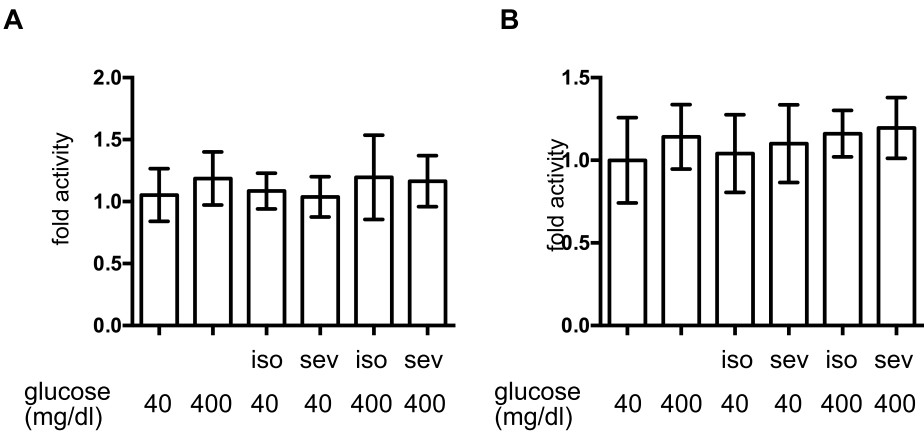

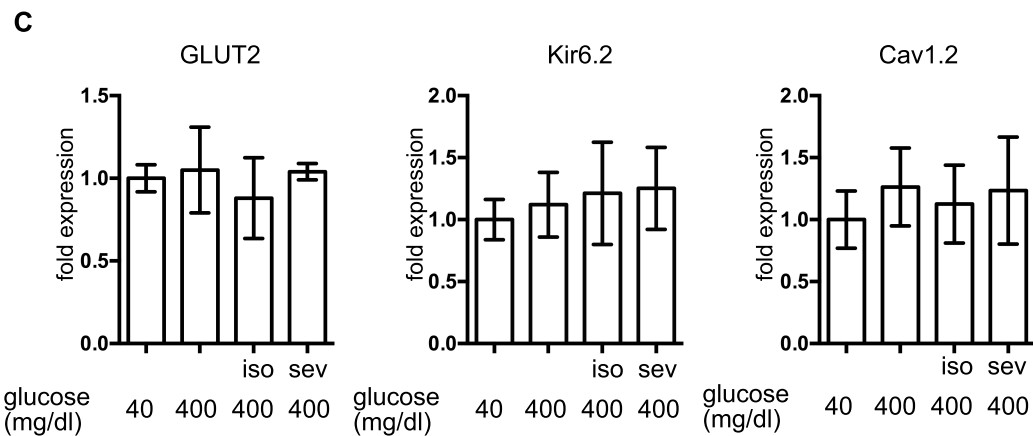

**Figure 3** **The effect of volatile anesthetics on molecular aspects of GSIS in MIN6 cells.** MIN6 cells were exposed to the indicated concentration of glucose and isoflurane (iso) or sevoflurane (sev) for 8 h. (A) Caspase 3/7 activity was assayed by the Apo-ONE$^{TM}$ Homogeneous Caspase-3/7 Assay (Promega, Madison, WI, USA) as described in 'Materials and Methods.' (B) Cell proliferation was assayed by colorimetric CellTiter 96$^{TM}$ AQueous One Solution Cell Proliferation Assay (Promega Corporation; Madison, WI) as described in 'Materials and Methods' ; this contains 3-(4,5-dimethylthiazol-2-yl)-5-(3-carboxymethoxyphenyl)-2-(4-sulfophenyl)-2H-tetrazolium, inner salt, and an electron coupling reagent (phenazine ethosulfate). (C) Cells were harvested and the mRNA levels of GLUT2, Cav1.2, and Kir6.2 were assayed by semi-quantitative reverse transcription real-time PCR. Data are presented as the mean $\pm$ SD ($n = 3$).

Neither 2.4% isoflurane nor 3.6% sevoflurane affected glibenclamide-induced insulin secretion (Fig. 4E).

## The effects of volatile anesthetics on the OCR

The elevated mitochondrial respiration observed in response to high-glucose simulation contributes to cellular hypoxia and HIF-1 activation in MIN6 cells (*Kurokawa et al., 2015*; *Sato et al., 2011*). We found that exposure to high levels of glucose significantly increased the OCR in MIN6 cells but to a lesser extent than the mitochondrial uncoupler, CCCP (5 μM) (Fig. 5). The mitochondrial electron transfer chain inhibitor, rotenone (100 nM),

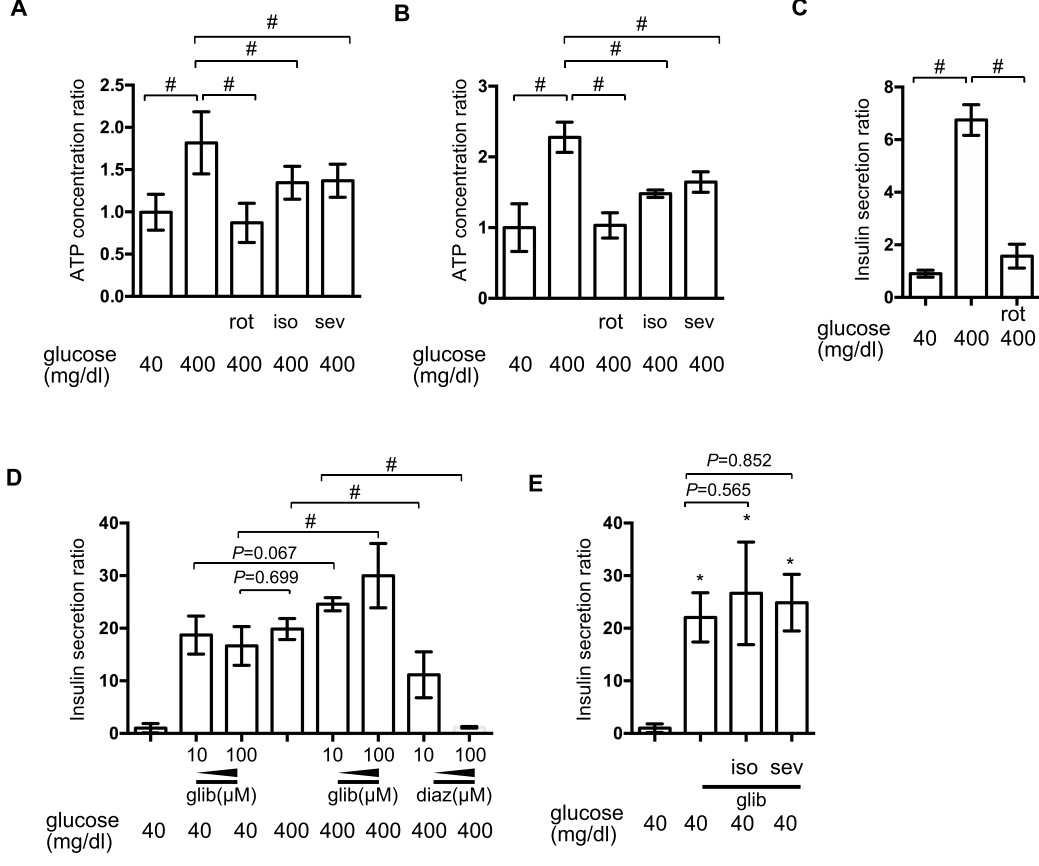

**Figure 4** **The effect of volatile anesthetics on intracellular signaling processes involved in GSIS.** (A) and (B) MIN6 cells were exposed to the indicated levels of glucose with rotenone (rot), 2.4% isoflurane (iso) (A), or 3.6% sevoflurane (sev) (B) for 2 h prior to analysis of the cellular ATP concentration, as described in 'Materials and Methods'. (C) and (D) MIN6 cells were exposed to the indicated levels of glucose with rotenone (C), diazoxide (diaz; D), or glibenclamide (glib; E) for 1 h prior to insulin determination. (E) MIN6 cells were exposed to 400 mg/dl glucose and glibenclamide with 2.4% isoflurane or 3.6% sevoflurane for 1 h prior to insulin determination. Data are presented as the mean ± SD (A and B: $N = 2$, $n = 6$, C: $N = 1$, $n = 3$, C and D: $N = 2$, $n = 6$); $*P < 0.05$ as compared with the control (no glibenclamide treatment); $\#P < 0.05$ for comparison of the indicated groups. $N$, number of independent experiments performed; $n$, number of samples.

suppressed the OCR. Importantly, 1.2% isoflurane and 1.8% sevoflurane also suppressed the OCR, although to a lesser extent than rotenone.

## The effects of volatile anesthetics on HIF-1 in GSIS

In order to elucidate the time course of HIF-1 activation, HIF-1$\alpha$ protein accumulation was investigated. As early as 1 h after exposure to 400 mg/dl glucose, HIF-1$\alpha$ protein accumulation was observed (Fig. 6A). Both of the tested volatile anesthetics significantly suppressed glucose-induced HIF-1$\alpha$ protein expression in MIN6 cells (Fig. 6B). Exposure to DMOG (100 μM) (*Zhou et al., 2007*) or nPG (100 μM) (*Kimura et al., 2008*) increased HIF-1$\alpha$ protein expression, which was not affected by isoflurane (Fig. 6C). The mRNA expression of HIF-1$\alpha$ was not affected by exposure to a high glucose level, volatile anesthetics, DMOG, or n-PG. In contrast, expression of downstream genes such as the

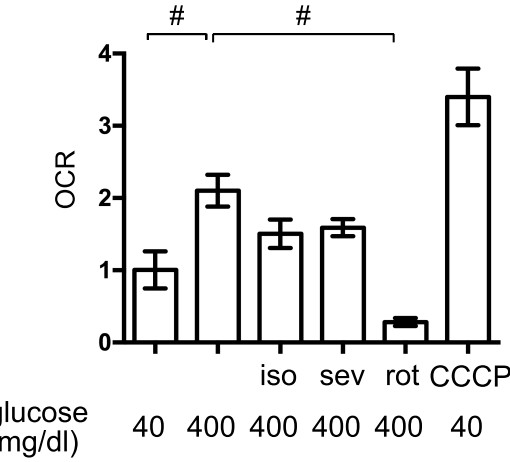

**Figure 5 The effects of volatile anesthetics on the OCR.** The oxygen consumption rate (OCR) of MIN6 cells was assayed under the indicated conditions, as described in 'Materials and Methods.' OCR was expressed as the ratio to that observed in MIN6 cells exposed to 40 mg/dl glucose. Data are presented as the mean ± SD ($n = 3$) $^*P < 0.05$, as compared with the control; #$P < 0.05$ for comparison of the indicated groups. iso, isoflurane; sev, sevoflurane; rot, rotenone; CCCP, carbonyl cyanide m-chlorophenylhydrazone.

glucose transporter 1 (GLUT1) and vascular endothelial growth factor (VEGF) were suppressed by isoflurane treatment. Importantly, this suppression was not observed by pretreatment with DMOG or n-PG (Fig. 6D).

## Impact of HIF-1 inhibition on GSIS

To elucidate the involvement of HIF-1 in glucose-elicited insulin secretion, effect of inhibition of HIF-1 activity was investigated by adopting siRNA against hif1$\alpha$ and the HIF-1$\alpha$ transcription inhibitor YC-1. Expression of mRNA of HIF-1$\alpha$ was suppressed by treatment with anti-hif1$\alpha$ siRNA (Fig. 7A) and induction of the expression of HIF-1$\alpha$ protein was also suppressed by anti-hif1$\alpha$ siRNA (Fig. 7B). YC-1 treatment did not affect the expression of mRNA and protein of HIF-1$\alpha$ subunit (Figs. 7A and 7B).

Inhibition of HIF-1 activity by YC-1 treatment and RNA interference-mediated knockdown of HIF-1$\alpha$ expression significantly suppressed GSIS in MIN6 cells (Fig. 7C). To examine if the constitutive activity of HIF-1 could rescue the suppression of GSIS by volatile anesthetics, MIN6 cells were incubated with 100 μM DMOG or nPG for 8 h before treatment with the volatile anesthetics. This inhibited HIF-$\alpha$ prolyl and asparaginyl hydroxylases, which induced HIF-1 activation, even under normoxic conditions. This pre-activation of HIF-1 inhibited the isoflurane-mediated suppression of GSIS and reduction of [ATPi] (Figs. 7D and 7E).

## DISCUSSION

This study demonstrated that the volatile anesthetics, isoflurane and sevoflurane, significantly suppressed GSIS in the mouse pancreatic $\beta$-cell-derived MIN6 cell line in clinically relevant doses. Our findings also indicated that these volatile anesthetics

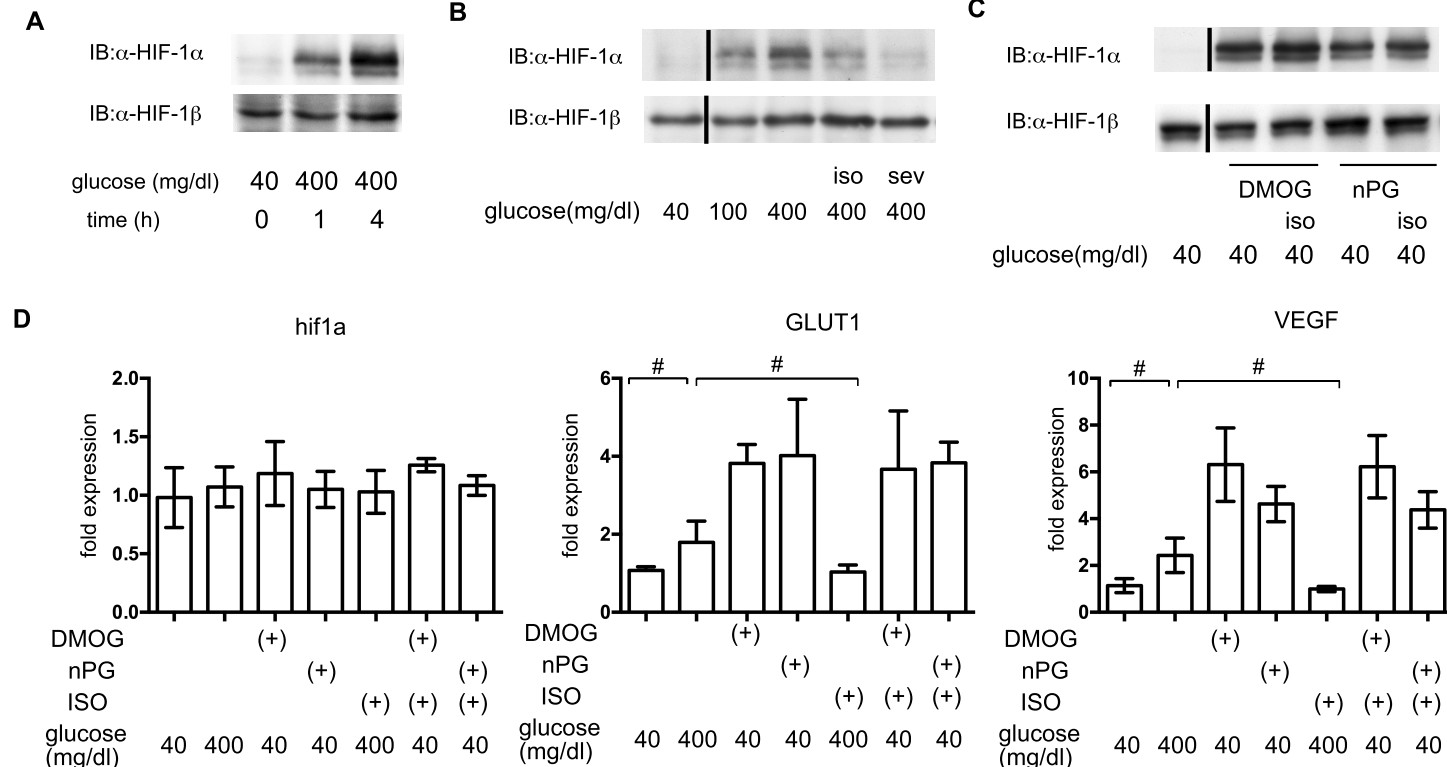

**Figure 6 The effects of volatile anesthetics on HIF-1 in GSIS.** (A), (B) and (C) These cells had been exposed to the indicated glucose levels, with or without isoflurane (iso) (B) or sevoflurane (sev) (B), or with dimethyloxaloylglycine (DMOG) or 3,4,5-trihydroxybenzoic acid propyl ester (n-PG), with or without isoflurane (C) with 20% $O_2$ for 4 h. MIN6 whole-cell lysates were immunoblotted (IB) twice to detect HIF-1$\alpha$ and HIF-1$\beta$ proteins and representative images are shown. (D) Cells were harvested for semi-quantitative real-time PCR for HIF-1$\alpha$, glucose transporter 1 (GLUT1), and vascular endothelial growth factor (VEGF). Cell cultures were repeated at least twice and the PCR analyses were performed in triplicate. Data are presented as the fold induction, relative to the level observed with 40 mg/dl glucose and no drug treatment. Data are presented as the mean $\pm$ SD ($n = 3$); #$P < 0.05$ for comparison of the indicated groups.

inhibited the glucose-induced increase in [ATPi] by suppressing HIF-1 activation in response to glucose-induced intracellular hypoxia.

Insulin secretion elicited by $K_{ATP}$ blockade using glibenclamide was not affected by either isoflurane or sevoflurane (Fig. 4E). In addition, we demonstrated that mRNA expression of GLUT2, Kir6.2 and Cav1.2 was not affected by isoflurane treatment (Fig. 3C). This evidence strongly suggests that the cellular processes involved in glucose intake or plasma membrane depolarization were not affected by these volatile anesthetics.

We also demonstrated that both isoflurane and sevoflurane inhibited the glucose-induced elevation of [ATPi] (Fig. 4A). This may be the most important mechanism underlying the GSIS inhibition induced by both of these anesthetics. HIF-1$\alpha$ protein expression is controlled by the cellular oxygen tension (*Hirota & Semenza, 2005*), which is reduced by exposure of cells to high-glucose conditions (*Kurokawa et al., 2015*; *Sato et al., 2011*). We previously reported that volatile anesthetics, including halothane, isoflurane and sevoflurane, inhibited hypoxia-induced HIF activation *in vitro and in vivo* (*Itoh et al., 2001*; *Kai et al., 2014*). Moreover, we demonstrated that the volatile anesthetics

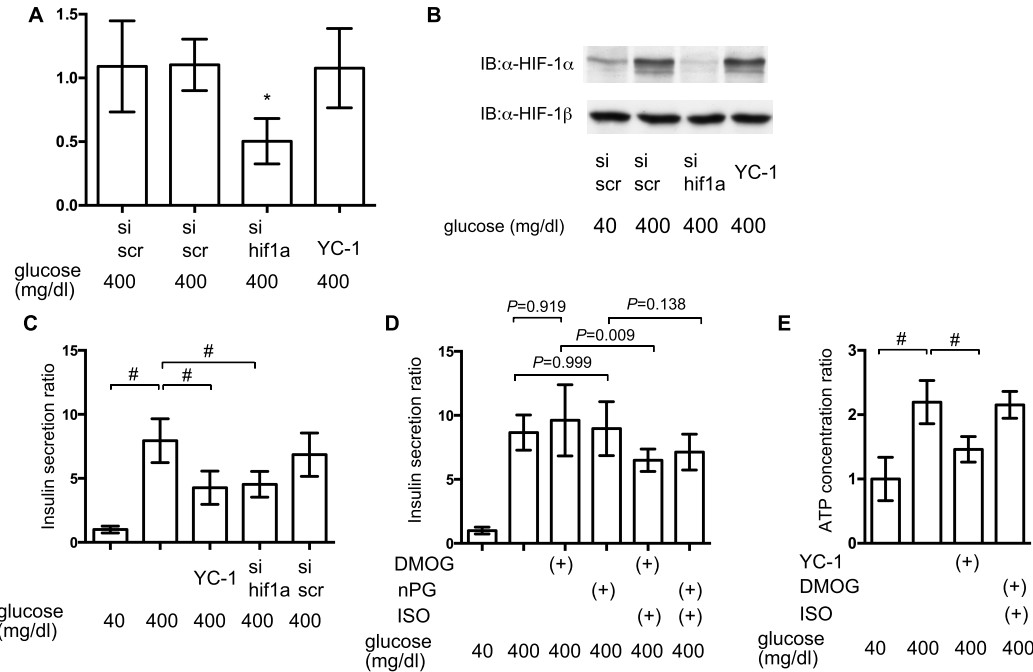

**Figure 7 Impact of HIF-1 inhibition on GSIS.** (A) and (B) MIN6 cells were transfected by 100 nM siRNA using HiPer-Fect™ Transfection Reagent (Qiagen, Valencia, CA, USA) or treated with YC-1. Cells were harvested for semi-quantitative real-time PCR for HIF-1$\alpha$ (A). Cells were harvested and whole-cell lysates were immunoblotted (IB) to detect HIF-1$\alpha$ and HIF-1$\beta$ proteins (B). (C) and (D) MIN6 cells were incubated with the indicated glucose concentrations for 1 h, with or without the indicated compounds, prior to determination of insulin secretion. Data are presented as the mean $\pm$ SD ($N = 2$, $n = 6$); #$P < 0.05$ for comparison of the indicated groups. (E) MIN6 cells were exposed to the indicated glucose concentrations and compounds for 2 h prior to analysis of the cellular ATP concentration, as described in 'Materials and Methods'. YC-1: 5-[1-(phenylmethyl)-1H-indazol-3-yl]-2-furanmethanol. Data are presented as the mean $\pm$ SD; #$P < 0.05$ for comparison of the indicated groups. $N$, number of independent experiments performed; $n$, number of samples.

partially attenuated glucose-induced oxygen consumption (Fig. 5). This may be one of the mechanisms by which volatile anesthetics suppress glucose-induced HIF-1 activation. Another novel finding of the present study was that glucose-induced ATP production was partially dependent on HIF-1 activity; YC-1-mediated inhibition of HIF-1 significantly reduced [ATPi] and GSIS (Fig. 7C). Moreover, DMOG- or nPG-induced pre-activation of HIF-1, which was resistant to volatile anesthetic treatment, attenuated the effects of volatile anesthetics on [ATPi] and GSIS induced by the volatile anesthetics (Figs. 7D and 7E). Our experimental results indicated that these volatile anesthetics also inhibited the increase in OCR (Fig. 5). In addition, glucose-induced the serine/threonine kinase AKT- and mammalian target of rapamycin (mTOR)-dependent HIF-1$\alpha$ signaling may partially contribute the HIF-1 activation observed in the present study (*Harada et al., 2009*; *Oda et al., 2006*). In a previous study, we reported that the volatile anesthetic halothane inhibited HIF-1$\alpha$ accumulation elicited by exposure to hypoxia and desferrioxamine in Hep3B cells (*Itoh et al., 2001*). In this study, however, HIF-1$\alpha$ accumulation by DMOG and nPG, which are identified as inhibitors of HIF-1$\alpha$-hydroxylases, are resistant to isoflurane treatment

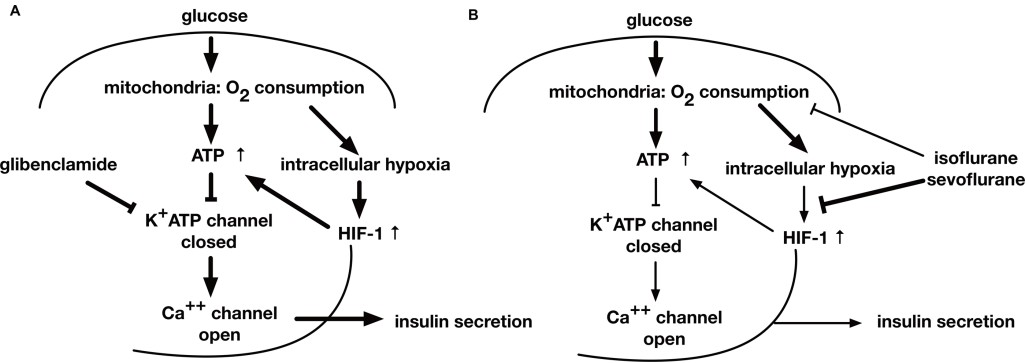

**Figure 8 A schematic diagram indicating the mechanism of GSIS suppression by volatile anesthetics.** (A) When the extracellular glucose concentration increases, pancreatic $\beta$-cell metabolism accelerates, leading to an increase in [ATPi]. As a result of these metabolic changes, the activity of ATP-dependent potassium channels ($K_{ATP}$) decreases; this causes membrane depolarization to the threshold potential at which voltage-dependent calcium channels open, allowing $Ca^{2+}$ influx. The ensuing increase in cytosolic $Ca^{2+}$ concentration triggers exocytosis of insulin-containing vesicles. (B) Glucose-induced oxygen consumption is partially suppressed by the volatile anesthetics (isoflurane and sevoflurane). Mitochondrial oxygen consumption leads to cellular hypoxia, which activates HIF-1. The process is also inhibited by volatile anesthetics. The glucose-induced increase in the intracellular ATP level is inhibited by volatile anesthetics. Consequently, these compounds suppress GSIS, which is largely dependent on the intracellular ATP concentration.

(Fig. 6C) in MIN6 cells. The reasons of this discrepancy are unknown at this moment. But the differences between halothane and isoflurane and between Hep3B cells and MIN6 cells may partially explains the discrepancy.

Although the involvement of HIF-1 in GSIS in pancreatic $\beta$-cells remains controversial because the role of this transcription factor in these cells is not fully understood, there is some evidence that HIF-1 influences insulin secretion. Islets were reported to be exposed to relatively hypoxic conditions, even under normal conditions (*Ashcroft, 2005*). Genetic ablation of HIF-1$\alpha$ or HIF-1$\beta$ significantly inhibited GSIS in mice (*Cheng et al., 2010*; *Gunton et al., 2005*; *Pillai et al., 2011*), while a forced artificial increase in HIF-1$\alpha$ protein and HIF-1 activity levels caused by genetic ablation of the von Hippel Lindau (VHL) protein in mouse pancreatic $\beta$-cells attenuated GSIS and increased lactate levels (*Cantley et al., 2009*; *Puri, Cano & Hebrok, 2009*; *Zehetner et al., 2008*). However, less than 3% of patients with VHL syndrome, caused by a loss of VHL function, showed abnormal glucose tolerance (*Ashcroft, 2005*). In addition, although mutations in subunits of the succinate dehydrogenase complex and in HIF-1$\alpha$ prolyl hydroxylases are associated with HIF-1 hyperactivity, there are no reported changes in glucose tolerance (*Ashcroft, 2005*). In HIF-1$\alpha$-deficient mice, the hematopoietic stem cells (HSCs) lost their cell cycle quiescence and HSC numbers decreased during various stressful conditions, including bone marrow transplantation and myelosuppression (*Takubo et al., 2010*). Conversely, increased HIF-1$\alpha$ protein levels in response to biallelic loss of VHL induced cell cycle quiescence in HSCs and their progenitors but impaired transplantation capacity (*Takubo et al., 2010*). This indicated that the fine tuning of HIF-1$\alpha$ protein expression and HIF-1 activity is critical in the regulation of biological responses. Moreover, the level of HIF-1$\alpha$ protein expression

influences insulin secretion; a mild increase of HIF-1$\alpha$ protein level is beneficial for $\beta$-cell function, whereas overexpression of HIF-1$\alpha$ protein, caused by VHL deletion or severe hypoxia, is deleterious for $\beta$-cell function (*Ashcroft, 2005*). Although the GSIS process is very rapid and we did not provide the results on the detail of molecular process between HIF-1 transcriptional activity and maintenance of intracellular ATP concentration, our experimental results indicate the involvement of HIF-1 in the suppressive effect of the volatile anesthetics of GSIS.

## CONCLUSION

This study examined the effects of the volatile anesthetics, sevoflurane and isoflurane, on GSIS. Both the anesthetics inhibited the glucose-induced increase of [ATPi], which is dependent on intracellular hypoxia-induced HIF-1 activity, and suppressed GSIS at a clinically relevant dose in MIN6 cells (Fig. 8).

## ACKNOWLEDGEMENTS

We would like to thank Dr. Hiroshi Harada at Kyoto University for critical reading of the manuscript, Dr. Miyazaki at Osaka University for gifting MIN6 and MIN7 cells and Editage (www.editage.jp) for English language editing.

### Funding

This work was supported by JSPS KAKENHI Grant Number 24659695 and 22659283 to KH, to 25462457 to KN, and 24592322 and 15K10551 to TA. The funders had no role in study design, data collection and analysis, decision to publish, or preparation of the manuscript.

### Grant Disclosures

The following grant information was disclosed by the authors:
JSPS KAKENHI: 24659695, 22659283, 25462457, 24592322, 15K10551.

### Competing Interests

The authors declare there are no competing interests.

### Author Contributions

- Kengo Suzuki and Kiichi Hirota conceived and designed the experiments, performed the experiments, analyzed the data, wrote the paper, prepared figures and/or tables, reviewed drafts of the paper.
- Yoshifumi Sato conceived and designed the experiments, wrote the paper, reviewed drafts of the paper.
- Shinichi Kai conceived and designed the experiments, performed the experiments, reviewed drafts of the paper.
- Kenichiro Nishi, Takehiko Adachi and Yoshiyuki Matsuo conceived and designed the experiments, reviewed drafts of the paper.

## Data Availability

Raw data is submitted as Supplemental Information 1.

## Supplemental Information

Supplemental information for this article can be found online at http://dx.doi.org/10.7717/peerj.1498#supplemental-information.

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
