# Peer review of "Volatile anesthetics suppress glucose-stimulated insulin secretion in MIN6 cells by inhibiting glucose-induced activation of hypoxia-inducible factor 1"

_PeerJ, doi:10.7717/peerj.1498_

## Round 0.1 · original submission · Major Revisions

This manuscript describes a clinically relevant , well designed and systematical research work. The following parts need clarification or amendment: statistical analysis, time points of measurements, discussion. Please also check the consistency of abbreviations/labeling.

·

Basic reporting

In the introduction, the authors state that "insulin secretion from pancreatic beta-cells in response to an increase in the blood glucose concentration plays the most critical role in glycemic control." This statement is not supported in the text by any literature references, so would be better stated as "plays a critical role." This revised statement would be generally accepted by those in the diabetes field as 'common knowledge'.

Experimental design

No comments

Validity of the findings

In the Materials and Methods section, it is stated that experiments were performed in triplicate on at least two separate occasions. This experimental design would only provide an N=2, which is not sufficient for statistical analysis. The results of at least 3 independent experimental replicates should be used for statistical analysis. In reading the figure legends, though, nearly all figures are stated to have an N=3-5. Thus, there is a discrepancy between what is stated in the materials and methods (which is potentially insufficient to draw conclusions from the data) and what is stated in the figure legends (which should be sufficient).

Also in the Materials and methods, it is stated that representative data are shown. It would be more convincing to show the means and SDs of the data combined from all 3-5 experimental replicates (presuming that the N values refer to experimental replicates and not technical replicates combined from 2 experiments).

In the legend for figure 5, the number of independent experimental replicates is not stated. This should be corrected.

In sum, the experimental design should be clarified throughout the manuscript to make it clear that any statistically significant differences are real and are not simply artifacts arising from an insufficient number of experimental replicates.

Additional comments

Overall, this manuscript described the results of well-designed experiments to answer an important question, with some clarification necessary to solidify the statistical analysis.

Reviewer 2 ·

Basic reporting

No Comments

Experimental design

Experiments were carried out systematically. However, in some experiments, time points of measurement were not adequate. For example, inhibition of insulin secretion induced by anesthetics was observed at 20 and 40 min. However, ATP was measured at 2 hr. Since the concentrations of ATP in response to glucose change considerably as a function of time, it should be measured at least within 40 min. Also, in Figures 6A and 6B, the expression of HIF-1 was measure at 4 hrs. Again, it should be measured within 1 hour to obtain meaningful conclusion. In experiments shown in Figure 6C, what is the time point of measurement?

Validity of the findings

1) Based on the results presented, it is clear that volatile anesthetics inhibit glucose metabolism and thereby elevation of ATP. But it is not totally clear whether volatile anesthetics reduced ATP by acting on HIF-1. The only set of data supporting the involvement of HIF-1 is the effect of YC-1, an inhibitor of HIF-1. In this regard, specificity of the action of YC-1 should be examined. Does YC-1 affect insulin secretion induced by high concentration of KCl? Also, it should be examined whether or not siHIF-1α reduces glucose-induced elevation of ATP as does YC-1 (Figure 6F). What is the efficiency of knockdown using siHIF-1α?

2) Do the authors really think HIF-1 controls glucose-induced elevation of ATP by a mechanism involving transcription? Note that changes in ATP occur within minutes and insulin secretion completes within 40 min. This question should be discussed in the Discussion.

Reviewer 3 ·

Basic reporting

No Comments

Experimental design

No Comments

Validity of the findings

No Comments

Additional comments

Volatile anesthetics is known to suppress insulin secretion by glucose stimulation (Tanaka 2011 Life Sci). They previous reported that intracellular hypoxia in pancreatic beta cells is induced by secretion of insulin by glucose stimulation along with induction of HIF1alpha (Sato 2011 JBC). In this paper, the authors show that volatile anesthetics suppressed increase of HIF1alpha and decreased intracellular ATP levels, resulting in decrease of insulin secretion. Linking the suppression of insulin secretion by volatile anesthetics to HIF1alpha and energy metabolism is a novel finding.

Comments
Western blot analysis for HIF1alpha after YC-1 or siHIF1alpha should be shown in Figure 6 to confirm the decrease of HIF1alpha in protein levels.

In a previous paper (Itoh 2001 FEBS Lett), it is reported that halothane suppresses HIF1alpha expression under hypoxic conditions as well as by CoCl2 or DFX treatment. In this manuscript, isoflurane did not suppress HIF1alpha expression by DMOG or nPG treatment. The discrepancy should be discussed.

Figure 7 (a schematic diagram) is not easy to understand. The diagram should be separated into two situations; with or without volatile anesthetics treatment, changing the thickness of the arrows.

Other comments
In line 115 (Materials and Methods), for measurement of OCR, the timing of the treatment of isoflurane, sevoflurane, or CCCP should be described more in detail.

In line 231, the sentence, ‘Importantly, this suppression was reversed by pretreatment with DMOG or n-PG (Fig. 6C)’ sounds strange as the experiments were performed under relatively low glucose conditions (40mg/mL). ‘not observed’ would be better than ‘reversed’.

In line 438 (Figure 1 legend), glucose (gluc) should be glucose (glc).

In line 453 (Figure 3 legend), 4hr (B) is described as 8-h exposure in the Results (line197), and 4h in the Methods (line 99). Those should be consistent.

In line 463 (Figure 4 legend), the title is for Figure 5.

In Figure 1C, glu should be glc.

---

## Round 0.2 · Minor Revisions

The manuscript has been greatly improved, only minor issues remained to be corrected. please carefully check the paper as suggested by reviewer 3.

Reviewer 2 ·

Basic reporting

No Comments

Experimental design

No Comments

Validity of the findings

NO Comments

Additional comments

The manuscript was revised according to the comments.

Reviewer 3 ·

Basic reporting

none

Experimental design

none

Validity of the findings

none

Additional comments

The author replied well to our comments.
Note that the corrections are not properly reflected to the text.

Text
Line 220 81h
Line 234 But the effects were weaker than 100nM rotenone, which suppressed GSIS (Fig. 4C). → the figure should be indicated here.
Line 238 In MIN6 cells, glibenclamide induced GSIS in the presence of 400 mg/dl
239 glucose, but even with 40 mg/dl glucose → Effect of glibenclamide is supposed to be through inhibition of K-ATP channel and nothing to do with glucose, so that it is not appropriate to call it as GSIS (glucose stimulated insulin secretion). ‘Insulin secretion’ would be better.

Line 240 Fig.4C should be Fig. 4D
Line 241 Fig.4D should be Fig. 4E
Line 284 Fig.4D should be Fig. 4E

Figure legend
Following are the errors the reviewer found. Please check the figure legend thoroughly.
Figure 4
Line 503 Description about B is missing.
Line 506 D. MIN6 cells… should be E
Line 507 400mg/dl should be 40mg/dl
Line 507 Description about sevoflurane is missing.

Figure 6
Line 523 The sentence is about (B) and that about (A) is missing.
Line 524 with or without isoflurane (B) → The sentence is about (C).
Line 525 C. Cells were harvested → The sentence is about (D).

---

## Round 0.3 · accepted · Accept

All comments were addressed.